# A 3-D Near-Field Source Localization Approach Based on the Combination of a Phase Interferometer, the Centroid Algorithm and the Perpendicular Foot Algorithm

**DOI:** 10.3390/s24196364

**Published:** 2024-09-30

**Authors:** Zhijun Qin, Tengfei Xie, Chen Xie, Ziwei Ma, Di He, Xin Chen, Wenxian Yu

**Affiliations:** 1Institute of Special Equipment Inspection and Research, Jiangxi General Institute of Testing and Certification, Nanchang 330029, China; 2Shanghai Key Laboratory of Navigation and Location-Based Services, Shanghai Jiao Tong University, 800 Dongchuan Road, Shanghai 200240, China

**Keywords:** near-field (NF) source, Root MUSIC, phase interferometer, centroid algorithm, perpendicular foot algorithm, 3-dimensional (3-D) localization

## Abstract

In this study, several 3-dimensional (3-D) parameter estimation and localization algorithms for wireless near-field (NF) sources are proposed employing the uniform circular array (UCA) structure. In the single-base-station case, the algebraic relation is demonstrated between the azimuth angle under the far-field (FF) assumption and the actual NF source firstly. Secondly, two groups of antenna pairs are selected with distances less than half the wavelength, which are called short baselines in the interferometer method. The foregoing short-baseline method is qualified to localize an NF source. In addition, a long-baseline method is also proposed with further research. Two groups of antenna pairs with distances greater than half the wavelength are selected as two long baselines. In the multiple-base-stations case, another two novel algorithms are also proposed. The first one is the centroid algorithm, which is based on the centroid calculation of three estimated source locations. And the second one is the perpendicular foot algorithm, which takes the perpendicular foot within three estimated source locations as the final positioning location. Simulation results illustrate that the proposed algorithms can achieve higher localization accuracy than the conventional 3-D Root MUSIC method. Moreover, the long-baseline method performs better than the short-baseline method. And it is also shown that the proposed perpendicular foot algorithm shows better performance than the proposed centroid algorithm.

## 1. Introduction

Source localization using an antenna array structure has been intensively addressed in wireless communication, radars, radio astronomy, sonar, etc. [1,2,3,4]. The rapid development of the sixth-generation (6G) communication technology provides convenience for the foregoing source localization [5,6], according to the fact that the key characteristic of 6G is to support amounts of antenna elements with a controllable direction at the transmitter and the receiver especially at the macrocell and microcell base stations, or even some picocell base stations. And in the indoor circumstance, the density of obstacles is high; however, the diffraction ability of the 6G signal may be poor. The cooperation of multiple base stations is generally necessary to achieve a high-precision positioning result.

The location information of a source contains three dimensions: the azimuth angle, the elevation angle, and the range. For the far-field (FF) sources, location information refers to the 2-D incident angle, which is a direction finding application. For the near-field (NF) sources, it is needed to obtain the 3-D position information within the range, which is a positioning application.

Various algorithms have been developed to deal with direction-of-arrival (DOA) estimation and the localization issue of sources, such as the multiple signal classification (MUSIC) algorithm [7,8,9,10], the ESPRIT (Estimation of Signal Parameters via Rotational Invariant Techniques) algorithm [11,12,13,14,15], and their derivatives. However, most of them are only applicable under the far-field assumption. In the near-field scenario, some algorithms are developed, such as the 3-D MUSIC algorithm [16], the path-following algorithm [17,18], and the higher-order ESPRIT algorithm [19]. However, most of these methods require amounts of computation and suffer from poor real-time ability, which also restrict their applications seriously.

In this study, we propose a solution to the 3-D parameter estimation and localization for a near-field source, by efficiently using the joint phase interferometer, the centroid algorithm and the perpendicular foot algorithm, which is also an extended work of our previous research [20]. The proposed methods are capable of yielding reasonably good estimates of the azimuth angle, the elevation angle, the range and then position results. In addition, the proposed approaches are of high efficiency in the sense that they do not require the 3-D spectrum search process. By comparing with other state-of-the-art algorithms, it can be found that the positioning error can significantly be reduced under the near-field condition. And when the proposed long-baseline phase interferometer technique and the proposed perpendicular foot technique are used, they may also introduce better performances other than the proposed short-baseline phase interferometer technique and the proposed centroid technique.

The remainder of this paper is arranged as follows. Firstly, the signal model and traditional wireless channel parameter estimation and localization methods are presented in Section 2. Then, in Section 3, near-field source localization approaches based on a single base station and multiple base stations are proposed, respectively. For the single-base-station condition, two methods based on a short-baseline phase interferometer and a long-baseline phase interferometer techniques are introduced. And for the multiple-base-stations condition, the centroid algorithm and the perpendicular foot algorithm are introduced, which also combine with the previous estimation based on the single-base-station condition. For the proposed approaches, the corresponding computer simulations are given and discussed in Section 4, which firstly evaluate the better localization performance of the proposed approaches overwhelming some other state-of-the-art algorithms, and secondly show the advantages of the long-baseline phase interferometer and the perpendicular foot algorithm. Finally, Section 5 concludes this paper.

## 2. The Near-Field Signal Model

In this study, without loss of generality and to simplify the corresponding analyses, we consider a UCA with M (M is an even number) identical omnidirectional sensors, with the geometry is presented in Figure 1. The circumference has the radius R. A near-field source is supposed to locate at range r measured from the origin of the UCA, the azimuth angle θ and the elevation angle φ.

Let the center of array as the phase reference origin, the array output can be expressed as
(1)$$X\left(t\right)=As\left(t\right)+n\left(t\right),$$
where $s\left(t\right)$ is a near-field signal, $X\left(t\right)$ is the array receiving vector which can be expressed as
(2)$$X\left(t\right)={\left[x_{0}\left(t\right),x_{1}\left(t\right),\cdots ,x_{M-1}\left(t\right)\right]}^{T},$$
where ${\left(\cdot \right)}^{T}$ is the transpose operator; $n\left(t\right)$ in (1) is the additive noise vector denoted by
(3)$$n\left(t\right)={\left[n_{0}\left(t\right),n_{1}\left(t\right),\cdots ,n_{M-1}\left(t\right)\right]}^{T},$$A in (1) denotes the M×1 dimension steering vector with
(4)$$A={\left[a_{0}\left(r,\theta ,\varphi \right),\cdots ,a_{k}\left(r,\theta ,\varphi \right),\cdots ,a_{M-1}\left(r,\theta ,\varphi \right)\right]}^{T},\left(k=0,1,\cdots ,M-1\right)$$
where $a_{k}\left(r,\theta ,\varphi \right)$ is expressed as
(5)$$a_{k}\left(r,\theta ,\varphi \right)=e^{-j\omega \tau_{k}\left(r,\theta ,\varphi \right)},$$
where $\tau_{k}\left(r,\theta ,\varphi \right)$ denotes the time difference of wave propagating from the source to the *k*th sensor which is given by
(6)$$\tau_{k}\left(r,\theta ,\varphi \right)=\frac{r-r_{k}\left(r,\theta ,\varphi \right)}{c_{0}},$$
where $c_{0}$ denotes the wave propagation speed, $r_{k}\left(r,\theta ,\varphi \right)$ represents the distance from the source to the *k*th antenna which is given by
(7)$$r_{k}\left(r,\theta ,\varphi \right)=\sqrt{r^{2}+R^{2}-2Rr\gamma_{k}},$$
where $\gamma_{k}=sin\varphi \cdot cos\left[\frac{2\pi /M}{k}-\theta \right]$. With a Fresnel approximation [21], $r_{k}\left(r,\theta ,\varphi \right)$ can be approximated by
(8)$$r_{k}\left(r,\theta ,\varphi \right)\approx r\left[1-\gamma_{k}\frac{R}{r}+\frac{1-\gamma_{k}^{2}}{2}{\left(\frac{R}{r}\right)}^{2}+\frac{\gamma_{k}-\gamma_{k}^{3}}{2}{\left(\frac{R}{r}\right)}^{3}\right].$$

Most conventional positioning methods are efficient only to the far-field source, while they perform badly in near-field source scenario. Further, many near-field applicable methods suffer from computation burden due to 3-D spectrum search or high-order statistics computation. In the rest of this paper, we propose several new ideas to estimate 3-D parameters for the near-field source under the following assumptions:(1)The wireless channel noise is an additive white Gaussian noise (AWGN) process which is statistically independent from the source signal.(2)The distance from adjacent sensors is assumed to be less than half of wavelength. For nonadjacent ones, the distance is greater than half of wavelength.(3)Assume that $\theta \in \left[0,2\pi \right]$ and $\varphi \in \left[0,\pi /2\right]$ to avoid ambiguity.

## 3. The Proposed Algorithm Based on a Joint Phase Interferometer, the Centroid Algorithm and the Perpendicular Foot Algorithm

In this section, we propose a novel algorithm for estimating 3-D parameters and realizing the wireless localization for near-field source. Firstly, we estimate the azimuth angle θ, the elevation angle φ and the range r, then according to Figure 1 the 3-D coordinates of the source can be given by
(9)$$\left(x,y,z\right)=\left(r\sin\varphi \cos\theta ,r\sin\varphi \sin\theta ,r\cos\varphi \right).$$

### 3.1. Near-Field Source Localization Based on a Single Base Station

#### 3.1.1. Estimate the Azimuth Angle of the Near-Field Source

When the receiver receives a signal, it can be assumed to be a near-field signal or a far-field signal. Define the azimuth and elevation of the assumed near-field source as $\tilde{\theta }$ and $\tilde{\varphi }$, respectively, and define those of the assumed far-field source as $\bar{\theta }$ and $\bar{\varphi }$, respectively. Then, the time delay of wave propagating from the source under the far-field assumption to the *k*th antenna and to the origin of the UCA is given by
(10)$$\tau_{k}\left(\bar{\theta },\bar{\varphi }\right)=\frac{1}{c_{0}}R\cdot sin\bar{\varphi }\cdot cos\left(\frac{2\pi }{M}k-\bar{\theta }\right).$$

Define a cost function which is given by
(11)$$L=\sum_{k=0}^{M-1}{\left[\tau_{k}\left(\bar{\theta },\bar{\varphi }\right)-\tau_{k}\left(r,\tilde{\theta },\tilde{\varphi }\right)\right]}^{2}\;=\sum_{k=0}^{M-1}\left[\frac{1}{c_{0}}R\cdot sin\bar{\varphi }\cdot cos\left(\frac{2\pi }{M}k-\bar{\theta }\right){\left.-\frac{1}{c_{0}}\left(r-r_{k}\left(r,\tilde{\theta },\tilde{\varphi }\right)\right)\right]}^{2}\right..$$

According to [22], the estimate $\left(\hat{\theta },\hat{\varphi }\right)$ of the far-field incident angles $\left(\bar{\theta },\bar{\varphi }\right)$ can minimize the above sum of squared errors of the time difference of (11) under the far-field assumption and that under the near-field assumption. By letting $\frac{\partial L}{\partial \tilde{\theta }}=0$, we get
(12)$$\tilde{\theta }=\hat{\theta }.$$

The result in (12) indicates that if the actual source is a near-field source, the estimate of the azimuth angle under the far-field assumption equals to the actual near-field azimuth angle, regardless of the range and the elevation angle. Based on this fact, we can estimate the azimuth angle of the assumed far-field source using the conventional MUSIC algorithm by seeking out the peak of the 2-D spatial spectrum of
(13)$$P\left(\hat{\theta },\hat{\varphi }\right)=\frac{1}{A_{0}^{H}\left(\hat{\theta },\hat{\varphi }\right)E_{N}E_{N}^{H}A_{0}\left(\hat{\theta },\hat{\varphi }\right)},$$
where $A_{0}\left(\hat{\theta },\hat{\varphi }\right)=A\left(\infty ,\hat{\theta },\hat{\varphi }\right)$, $E_{N}$ is the noise subspace of the array output covariance matrix, and ${\left(\cdot \right)}^{H}$ denotes complex conjugate transformation operator.

According to the above calculation, the estimated azimuth angle of the near-field source $\hat{\theta }$ can then be estimated unbiasedly.

#### 3.1.2. Estimate the Elevation Angle and Range Parameters of NF Source Based on a Short-Baseline Phase Interferometer

The widely used phase-interferometer-based algorithms exploit the received phase information by the array to estimate parameters [23,24]. In the actual measurement process, suppose the phase obtained by the phase detector is within the range $\left(-\pi ,+\pi \right)$. In the interferometer-based algorithm, when the distance between a group of sensor pairs is less than half of wavelength, it is defined as a short baseline, the measured phase difference is exactly true value. Otherwise, it is a long baseline, which may suffer from phase ambiguity. According to [17], the longer the baseline is, the smaller the estimation error will be.

In this step, we take advantage of two short baselines to estimate the elevation angle and the range of the near-field source. Considering that the adjacent sensors spacing is less than half of wavelength, we choose the sensor 0 and sensor 1, the sensor 0 and the sensor M−1 as two groups of short baselines, which correspond to line *a* and line *b*, respectively, in Figure 2. The phase differences measured by a phase detector are $\phi_{a}^{’}$ and $\phi_{b}^{’}$, respectively.

The distance between the signal arriving at the origin and the signal arriving at the sensor 0, the sensor 1, the sensor M−1 can be expressed separately as
(14)$$r_{0}\left(r,\varphi \right)=\sqrt{r^{2}+R^{2}-2Rr\cdot sin\varphi \cdot cos\hat{\theta }},$$
(15)$$r_{1}\left(r,\varphi \right)=\sqrt{r^{2}+R^{2}-2Rr\cdot sin\varphi \cdot cos\left(\frac{2\pi }{M}-\hat{\theta }\right)},$$
(16)$$r_{M-1}\left(r,\varphi \right)=\sqrt{r^{2}+R^{2}-2Rr\cdot sin\varphi \cdot cos\left[\frac{2\pi }{M}\left(M-1\right)-\hat{\theta }\right]}.$$

The phase difference between antenna 0 and antenna 1 is given by
(17)$$\phi_{a}=\frac{1}{c_{0}}2\pi f\left[r_{0}\left(r,\varphi \right)-r_{1}\left(r,\varphi \right)\right],$$
where f denotes the signal frequency.

Similarly, the phase difference between and sensor 0 and sensor M−1 is given by
(18)$$\phi_{b}=\frac{1}{c_{0}}2\pi f\left[r_{0}\left(r,\varphi \right)-r_{M-1}\left(r,\varphi \right)\right],$$

By letting $\phi_{a}^{’}=\phi_{a}$ and $\phi_{b}^{’}=\phi_{b}$, the algebraic relation between the elevation angle φ, range r and $\phi_{a}$, $\phi_{b}$ can be given by
(19)$$\phi_{a}^{’}=\frac{1}{c_{0}}2\pi f\left[r_{0}\left(r,\varphi \right)-r_{1}\left(r,\varphi \right)\right],$$
(20)$$\phi_{b}^{’}=\frac{1}{c_{0}}2\pi f\left[r_{0}\left(r,\varphi \right)-r_{M-1}\left(r,\varphi \right)\right].$$

Solve (19), (20) and we can obtain the estimates of φ and r. Then, the positioning result can be calculated according to (9). Based on the above proposed short baselines method, we can estimate 3-D parameters and realize the wireless localization for the near-field source. Considering the estimation error gets smaller as the baseline gets longer, we try to employ two long baselines to achieve a better performance in next step.

#### 3.1.3. Estimate the Elevation Angle and Range Parameters of NF Source Based on a Long-Baseline Phase Interferometer

In this step, we change to use two group of long baselines. Obviously, it is needed to solve ambiguity. Choose the sensor 0 and sensor $\frac{M}{2}$, the sensor 0 and the sensor $\frac{M}{2}-1$ as two groups of long baselines, which correspond to line *c* and line d, respectively, in Figure 2.

Similar to (17) and (18), the phase difference of baseline *c* and baseline *d* are given by
(21)$$\phi_{c}=\frac{1}{c_{0}}2\pi f\left[r_{0}\left(r,\varphi \right)-r_{M/2}\left(r,\varphi \right)\right],$$
(22)$$\phi_{d}=\frac{1}{c_{0}}2\pi f[r_{0}\left(r,\varphi \right)-r_{\frac{M}{2}-1}\left(r,\varphi \right)].$$

The phase difference between the mth sensor and the nth sensor is given by
(23)$$\phi_{m}-\phi_{n}=\frac{2\pi }{\lambda }\left[r_{m}\left(r,\theta ,\varphi \right)-r_{n}\left(r,\theta ,\varphi \right)\right].$$

Combining with (8), we can get
(24)$$\phi_{m}-\phi_{n}=\frac{2\pi }{\lambda }\left[R\left(\gamma_{n}-\gamma_{m}\right)+\frac{R^{2}}{2r}\left(\gamma_{n}^{2}-\gamma_{m}^{2}\right)+\frac{R^{3}}{2r^{2}}\left({\gamma_{m}-\gamma_{n}+\gamma }_{n}^{3}-\gamma_{m}^{3}\right)\right],$$
where $\gamma_{k}=sin\varphi \cdot cos\left(\frac{2\pi /M}{k}-\theta \right)$ and k=m,n.

When the baseline consisting of the mth sensor and the nth sensor is a long baseline, there is
(25)$$\phi_{m}-\phi_{n}=\phi_{dec}+2k_{0}\pi ,\;\left(k_{0}\in N\right)$$
where $k_{0}$ denotes the ambiguity number and $\phi_{dec}$ is the phase difference measured by a phase detector. Because $\phi_{dec}\in \left[-\pi ,+\pi \right]$, we can get
(26)$$-\pi {\leq \phi }_{m}-\phi_{n}-2k_{0}\pi \leq \pi ,$$
(27)$$-\frac{1}{2}-\frac{\phi_{m}-\phi_{n}}{2\pi }\leq k_{0}\leq \frac{1}{2}+\frac{\phi_{m}-\phi_{n}}{2\pi }.$$

Because we need to go through all possible values of $k_{0}$, its range can be given by
(28)$$-\frac{1}{2}-\frac{{\left(\phi_{m}-\phi_{n}\right)}_{max}}{2\pi }\leq k_{0}\leq \frac{1}{2}+\frac{{\left(\phi_{m}-\phi_{n}\right)}_{max}}{2\pi }.$$

For $\gamma_{k}\in \left[-1,+1\right]$, we have
(29)$${\left(\gamma_{n}-\gamma_{m}\right)}_{max}=2,$$
(30)$${\left(\gamma_{n}^{2}-\gamma_{m}^{2}\right)}_{max}=1,$$
(31)$${\left(\gamma_{m}-\gamma_{n}+\gamma_{n}^{3}-\gamma_{m}^{3}\right)}_{max}=4.$$

Substitute (29)–(31) into (24), we can get
(32)$${\left(\phi_{m}-\phi_{n}\right)}_{max}=\frac{2\pi }{\lambda }\left(2R+\frac{R^{2}}{2r}+\frac{R^{3}}{2r^{2}}\right).$$

Substitute (32) into (28), then we have
(33)$$-\frac{1}{2}-\frac{1}{\lambda }\left(2R+\frac{R^{2}}{2r}+\frac{R^{3}}{2r^{2}}\right)\leq k_{0}\leq \frac{1}{2}+\frac{1}{\lambda }\left(2R+\frac{R^{2}}{2r}+\frac{R^{3}}{2r^{2}}\right).\;\left(k_{0}\in N\right)$$

The near-field source is in Fresnel region as
(34)$$r\in \left[0.62{\left(\frac{D^{3}}{\lambda }\right)}^{\frac{1}{2}},\frac{{2D}^{2}}{\lambda }\right],$$
where D denotes the array aperture.

Apply the minimum value of r to (33), all possible values of the ambiguity number $k_{0}$ are obtained. Knowing measured phase differences and using (25), all possible combinations of phase differences of baseline c and baseline d are obtained. The corresponding set of possible values are marked as $\phi_{c}^{’}$ and $\phi_{d}^{’}$, respectively.

By letting $\phi_{c}^{’}=\phi_{c}$ and $\phi_{d}^{’}=\phi_{d}$, the algebraic relation between the elevation angle φ, range r and $\phi_{c}^{’}$, $\phi_{d}^{’}$ can be given by
(35)$$\phi_{c}^{’}=\frac{1}{c_{0}}2\pi f\left[r_{0}\left(r,\varphi \right)-r_{M/2}\left(r,\varphi \right)\right],$$
(36)$$\phi_{d}^{’}=\frac{1}{c_{0}}2\pi f[r_{0}\left(r,\varphi \right)-r_{\frac{M}{2}-1}\left(r,\varphi \right)].$$

Solve (35), (36) and we can obtain the estimates of φ and r. Considering that there are some combinations of possible phase values, we can obtain some group of solutions of φ and r accordingly. Put all sets of φ and r to (17) and (18), and we obtain some data of phase differences of baseline *a* and baseline *b*, marked as $\phi_{a}^{”}$ and $\phi_{b}^{”}$, respectively. The correlation calculation between the obtained phase differences $\phi_{a}^{”}$, $\phi_{b}^{”}$ and the measured phase differences $\phi_{a}$, $\phi_{b}$ is given by
(37)$$R_{\phi \phi }=\sqrt{{\left(\phi_{a}-\phi_{a}^{”}\right)}^{2}+{\left(\phi_{b}-\phi_{b}^{”}\right)}^{2}}.$$

When $R_{\phi \phi }$ reaches the minimum value, the similarity between the corresponding phase differences and the measured actual phase differences is at the maximum value. Therefore, the corresponding ambiguity number will equal to the real value, and the corresponding estimated elevation angle and range can be achieved. And based on (9), we can obtain the positioning result of the near-field source.

The influence of the range r and its estimation error on the positioning error are evaluated as follows. The root mean square error (RMSE) is used to evaluate the performance and it is defined as
(38)$$RMSE=\sqrt{\frac{1}{N_{s}}\sum_{i=1}^{N_{s}}\left[{\left(\hat{x}-x\right)}^{2}+{\left(\hat{y}-y\right)}^{2}+{\left(\hat{z}-z\right)}^{2}\right]},$$
where $\hat{x}$, $\hat{y}$, $\hat{z}$ are the coordinates calculated by the estimated parameters using (9), and x, y, z are the real coordinates of the source.

Considering the estimation errors of the range r, the azimuth θ and the elevation φ, the estimated coordinates can be given by
(39)$$\hat{x}=\left(r+\Delta r\right)\cdot sin\left(\varphi +\Delta \varphi \right)\cdot cos\left(\theta +\Delta \theta \right),$$
(40)$$\hat{y}=\left(r+\Delta r\right)\cdot sin\left(\varphi +\Delta \varphi \right)\cdot sin\left(\theta +\Delta \theta \right),$$
(41)$$\hat{z}=\left(r+\Delta r\right)\cdot cos\left(\varphi +\Delta \varphi \right),$$
where Δr, Δφ and Δθ denote the estimation error of r, φ and θ, respectively. Substitute (39)–(41) into (38), the RMSE can be given by
(42)$$RMSE=\sqrt{\frac{1}{N_{s}}\sum_{i=1}^{N_{s}}G\left(r,\theta ,\varphi ,\Delta r,\Delta \theta ,\Delta \varphi \right)},$$
where $G\left(r,\theta ,\varphi ,\Delta r,\Delta \theta ,\Delta \varphi \right)$ can be given by
(43)$$G\left(r,\theta ,\varphi ,\Delta r,\Delta \theta ,\Delta \varphi \right)=r^{2}G_{1}+2r\Delta rG_{2}+\Delta^{2}rG_{3},$$
where $G_{1}$, $G_{2}$ and $G_{3}$ can be given by
(44)$$G_{1}\left(r,\theta ,\varphi ,\Delta r,\Delta \theta ,\Delta \varphi \right)\;={sin}^{2}\varphi {\cdot cos}^{2}\theta +{sin}^{2}\left(\varphi +∆\varphi \right){\cdot cos}^{2}\left(\theta +∆\theta \right)\;-2sin\varphi \cdot cos\theta \cdot sin\left(\varphi +∆\varphi \right)\cdot cos\left(\theta +∆\theta \right),$$
(45)$$G_{2}\left(r,\theta ,\varphi ,\Delta r,\Delta \theta ,\Delta \varphi \right)\;={sin}^{2}\left(\varphi +∆\varphi \right){\cdot cos}^{2}\left(\theta +∆\theta \right)\;-sin\varphi \cdot cos\theta \cdot sin\left(\varphi +∆\varphi \right)\cdot cos\left(\theta +∆\theta \right),$$
(46)$$G_{3}\left(r,\theta ,\varphi ,\Delta r,\Delta \theta ,\Delta \varphi \right)={sin}^{2}\left(\varphi +∆\varphi \right){\cdot cos}^{2}\left(\theta +∆\theta \right).$$

The result in (12) indicates that if the actual source is an near-field source, the estimate of the azimuth angle under the far-field assumption equals to the actual near-field azimuth angle, regardless of the elevation angle and the range. we can estimate the azimuth angle of the assumed far-field source using the conventional Root MUSIC algorithm. According to [25], when the signal-to-noise ratio (SNR) is no less than 0 dB, the azimuth estimation of Root MUSIC algorithm can achieve high accuracy that is Δθ→0. In the near-field scenario, the SNR will not be too small. So the azimuth estimation of the proposed methods can also be precise. Then, $G_{1}$ and $G_{2}$ can be simplified as
(47)$$G_{1}={cos}^{2}\theta \cdot {\left[sin\varphi -sin\left(\varphi +∆\varphi \right)\right]}^{2},$$
(48)$$G_{2}={cos}^{2}\theta \cdot sin\left(\varphi +∆\varphi \right)\cdot \left[sin\left(\varphi +∆\varphi \right)-sin\varphi \right].$$

From (47) and (48), we can come to a conclusion that if ∆φ→0, it may lead to $G_{1}\to 0$ and $G_{2}\to 0$. Therefore, the RMSE can be given by
(49)$$RMSE=\sqrt{\frac{1}{N_{s}}\sum_{i=1}^{N_{s}}\left[{∆}^{2}r{sin}^{2}\left(\varphi +∆\varphi \right){cos}^{2}\left(\theta +∆\theta \right)\right]}.$$

From (49), it can be seen that the RMSE is uncorrelated to r.

Regarding the computational complexity, we compare the major multiplications involved in statistics matrix construction, EVD (Eigen-Value Decomposition) implementation and MUSIC spectrum search. The 3-D Root MUSIC method [16] constructs an M×M dimension matrix, perform its EVD, and executes DOA and range search, so the resulting multiplications are in order of
(50)$$O\left(NM^{2}+M^{3}+N_{\theta }N_{\varphi }N_{r}\left(M^{3}+1\right)\right),$$
where N is the snapshot number, M is the amount of array elements and $N_{\theta }$, $N_{\varphi }$, $N_{r}$ denote the numbers of search points in the dimensions of θ, φ, r, respectively. However, computational complexity of the proposed algorithms is given by
(51)$$O\left(NM^{2}+M^{3}+N_{\theta }N_{\varphi }\left(M^{3}+1\right)\right).$$

Therefore, it can be clearly seen that the proposed algorithms have lower computational cost than that of the 3-D Root MUSIC algorithm.

### 3.2. Near-Field Source Localization Based on Multiple Base Stations

In the following analysis, without loss of generality, it is supposed that the source signal is surrounded by multiple base stations, while it is in the near-field range of one base station, and may be regarded as far-field source related to other base stations, as shown in Figure 3. And for the far-field base stations, there are some line-of-sight (LOS) signals that have been blocked or they can only receive the reflection signal or non-line-of-sight (NLOS) signal.

#### 3.2.1. Near-Field Source Signal Localization Based on the Centroid Algorithm

A multi-station source signal localization based on the centroid algorithm is proposed in this subsection. Firstly, it is necessary to determine whether the source signal is a near-field source or a far-field source relative to each base station. If it is a near-field source, the long baseline-based near-field source localization algorithm proposed in Section 3.1.3 can be used, while if it is a far-field source, the traditional Root MUSIC algorithm can be used. Finally, the data from each base station can be fused to realize the signal source localization objective. To simplify the description and analyses, we take the scenario in Figure 3 as an example which can be regarded as the projection of the source signal and all base stations in the 2-D plane. Assuming that it can be determined that the LOS signal to base station D is obstructed by some obstacle, so the receiving signal at D will no longer be used, and only the LOS signal to base stations A, B, and C will be used for subsequent positioning calculation, with their 2-D coordinates recorded as $\left(x_{A},y_{A}\right)$, $\left(x_{B},y_{C}\right)$ and $\left(x_{C},y_{C}\right)$, respectively. Supposing $\left(x_{0},y_{0}\right)$ be the circle center of the minimum covering circle of the above three points in the 2-D plane, calculate the distances from $\left(x_{0},y_{0}\right)$ to three base stations A, B and C, as $d_{A}$, $d_{B}$ and $d_{C}$, respectively. The right boundary value of the Fresnel domain is denoted as $d_{0}=\frac{2D^{2}}{\lambda }$, where *D* is the array aperture and λ is the signal wavelength. Then, by comparing the sizes of $d_{A}$, $d_{B}$ and $d_{C}$, with $d_{0}$, it can determine whether the source is within or outside the Fresnel domain of corresponding base station in 2-D plane. Afterwards, there are two situations need to discuss: firstly, the source signal is outside the Fresnel domains of all base stations; and secondly, the source signal is within the Fresnel domain of one of the base stations and outside the Fresnel domains of other two base stations.

Let us discuss the first scenario first. The distance from the projection point of the source signal in 2-D plane to the base station must be smaller than the distance from the actual location of the source to the base station. Therefore, if the projection point of the source signal is outside the Fresnel domain of all base stations, then the source signal must be a far-field source relative to each base station. In this case, three base stations A, B and C use the Root MUSIC algorithm to estimate the incident angle of the far-field source, and it can establish the following spatial spectra:(52)$$P_{MUSIC}=\frac{1}{a^{H}\left({\hat{\theta }}_{i},\varphi \right){\hat{U}}_{Ni}{\hat{U}}_{Ni}^{H}a\left({\hat{\theta }}_{i},\varphi \right)}\;\left(i=A,B,C\right)$$
where ${\hat{\theta }}_{i}$ $\left(i=A,B,C\right)$ are the estimated azimuth angles of three base stations A, B and C, based on the projection point of the source signal, ${\hat{U}}_{Ni}$ $\left(i=A,B,C\right)$ are the estimated noise subspace obtained by feature decomposition of the covariance matrix of data received by each base station. So the two-dimensional search of the conventional MUSIC algorithm can be simplified to only one-dimensional search, which can greatly reduce the computational complexity. Suppose the estimated elevation angles of each base station to the source as ${\hat{\varphi }}_{A}$, ${\hat{\varphi }}_{B}$ and ${\hat{\varphi }}_{C}$, respectively. Next, we will introduce how to estimate the location of the source signal based on the parameters ${\hat{\theta }}_{A}$, ${\hat{\theta }}_{B}$, ${\hat{\theta }}_{C}$, ${\hat{\varphi }}_{A}$, ${\hat{\varphi }}_{B}$ and ${\hat{\varphi }}_{C}$.

Figure 4 shows the location of the source signal in 3-D space. *P* is the projection point of the source signal at 2-D plane xoy. The coordinate of the source is $\left(x,y,z\right)$. According to the spatial geometric relationship, the following relationship can be obtained between the coordinate of source *S* and the coordinates of base stations A, B, and C:(53)$$\left\{\begin{array}{c} x\cdot tan\theta_{A}-y=0\\ y-z\cdot tan\varphi_{A}\cdot sin\theta_{A}=0\\ \begin{array}{c} \begin{array}{c} \left(x-x_{B}\right)tan\theta_{B}-y=0\\ y-z\cdot tan\varphi_{B}\cdot sin\theta_{B}=0 \end{array}\\ \left(x_{C}-x\right)tan\theta_{C}+y=y_{C}\\ y-z\cdot tan\varphi_{C}\cdot sin\theta_{C}=y_{C} \end{array} \end{array}\right.$$

It can also be rewritten as the matrix form as
(54)HX=Z,
where $X={\left(x,y,z\right)}^{T}$, $Z={\left(0,0,x_{B}tan\theta_{B},0,y_{C}-x_{C}tan\theta_{C},y_{C}\right)}^{T}$. The matrix H can be written as
(55)$$H=\left[\begin{array}{ccc} tan\theta_{A} & -1 & 0\\ 0 & 1 & -tan\varphi_{A}{sin\theta }_{A}\\ tan\theta_{B} & -1 & 0\\ 0 & -1 & -tan\varphi_{B}{sin\theta }_{B}\\ -tan\varphi_{A}{sin\theta }_{C} & 1 & 0\\ 0 & 1 & -tan\varphi_{C}{sin\theta }_{C} \end{array}\right].$$

The optimal solution can then be obtained by the least squares method as $X={\left(H^{T}H\right)}^{-1}H^{T}Z$, and the localization estimation of the source signal can be acquired.

Next, we will focus on the second scenario: the projection point estimated (the center of the minimum coverage circle of three base stations) is within the Fresnel domain of one base station and outside the Fresnel domains of other two base stations. To simplify the description, it is assumed that the projection point is within the Fresnel domain of base station C and outside the Fresnel domains of base stations A and B. That is, the source signal must be a far-field source relative to base stations A and B, and for base station C, it may be a near-field source or a far-field source. The Root MUSIC algorithm can be used to estimate the incident angle of the source signal at base stations A and B, where ${\hat{\theta }}_{i}$ $\left(i=A,B\right)$ are the azimuth angle estimates of base stations A and B, and ${\hat{\varphi }}_{i}$ $\left(i=A,B\right)$ are the elevation angle estimates.

Based on the coordinates of base stations A and B, as well as the estimated results of two sets of azimuth and elevation angles, the equations for two target observation lines can be obtained as
(56)$$\left\{\begin{array}{c} x\cdot tan\theta_{A}-y=0\\ y-z\cdot tan\varphi_{A}\cdot sin\theta_{A}=0 \end{array}\right.$$
(57)$$\left\{\begin{array}{c} \left(x-x_{B}\right)\cdot tan\theta_{B}-y=0\\ y-z\cdot tan\varphi_{B}\cdot sin\theta_{B}=0 \end{array}\right.$$

In the ideal situation where there is no measurement error or noise impact, the two target observation lines must intersect at the actual position of the source signal. However, in practical situations, there are measurement errors and channel noise, so it is generally impossible for the target observation lines to intersect. Moreover, because the estimated values of the same source by the two base stations are not significantly different in theory, the two closest points on these two target observation lines can be used as the estimation of the target position by the base stations A and B, respectively. Therefore, it is necessary to find the common perpendicular of two straight lines and find the coordinates of the two perpendicular feet.

The process of finding a common perpendicular is a spatial geometric problem, which will not be detailed here. Suppose the perpendicular feet corresponding to base stations A and B are $P_{1}\left(x_{1},y_{1},z_{1}\right)$ and $P_{2}\left(x_{2},y_{2},z_{2}\right)$, respectively. Use these two perpendicular feet as the original estimates of the source position related to base stations A and B. The estimation of far-field sources by a single base station could only estimate the azimuth angles and elevation angles, thereby determining the direction of incoming waves. Now, the specific source location has been basically determined through the measurement signals of the two base stations. However, in order to further improve the positioning accuracy, information from base station C can also be introduced for calculation and estimation.

As mentioned before, the projection point is within the Fresnel domain of base station C, and the distance from the projection point to the base station C must be smaller than the distance from the actual position of the source to the base station C. Therefore, the true position of the source may be within or outside the Fresnel domain of base station C. The method to determine whether the source is a near-field source or a far-field source for base station C is: Calculate the distance from the point on the line $P_{1}P_{2}$ to base station C. When the angle estimation accuracy from the base station is relatively high, $P_{1}P_{2}$ should be theoretically a line with very short length. If the distance from points on the line $P_{1}P_{2}$ to base station C is greater than $\frac{2D^{2}}{\lambda }$, it can be determined that the source is a far-field source relative to base station C, while the least squares algorithm from (53) to (55) can determine the estimated position of the signal source after collecting all the data from three base stations. If the distance from points on the line $P_{1}P_{2}$ to base station C is less than $\frac{2D^{2}}{\lambda }$, it can be determined that it is a near-field source relative to base station C. The 3-D parameters of the source can be estimated according to the algorithm based on a long-baseline interferometer described before, and then the position of the source can be calculated and expressed as $P_{3}\left(x_{3},y_{3},z_{3}\right)$.

Now we have obtained three coordinate estimates for the signal source, which are $P_{1}\left(x_{1},y_{1},z_{1}\right)$, $P_{2}\left(x_{2},y_{2},z_{2}\right)$ and $P_{3}\left(x_{3},y_{3},z_{3}\right)$. Generally, they could form a triangle in 3-D space, and the centroid of the triangle can be taken as the final estimated position of the source by multiple base stations. That is, the estimated coordinate value of the signal source can be written as $P_{0}\left(x,y,z\right)$, where
(58)$$\left\{\begin{array}{c} x=\frac{x_{1}+x_{2}+x_{3}}{3}\\ y=\frac{y_{1}+y_{2}+y_{3}}{3}\\ z=\frac{z_{1}+z_{2}+z_{3}}{3} \end{array}\right.$$

In the above proposed algorithm, the centroid of three estimated source coordinates is taken as the final positioning value, so the proposed algorithm is called the signal source positioning method based on the centroid algorithm.

#### 3.2.2. Near-Field Source Signal Location Based on the Perpendicular Foot Algorithm

The source signal localization based on the perpendicular foot algorithm proposed in this subsection has a certain correlation with the algorithm proposed in the previous subsection. When the signal source is a far-field source for base stations A and B, find the common perpendicular lines of the two target observation lines, and use the two perpendicular lines as the estimated values of the signal source position for base stations A and B, respectively, which are denoted by $P_{1}\left(x_{1},y_{1},z_{1}\right)$ and $P_{2}\left(x_{2},y_{2},z_{2}\right)$, respectively. When the signal source is a near-field source for base station C, the algorithm based on a long baseline interferometer proposed in Section 3.1.3 is used to estimate the position of the source signal, which is denoted by $P_{3}\left(x_{3},y_{3},z_{3}\right)$.

The difference based on the perpendicular foot algorithm is that it uses the obtained perpendicular foot as the final estimation value for the positioning of the target signal source by multiple base stations. Assuming the final coordinate estimation value of the source signal is $P_{0}\left(x,y,z\right)$, based on the condition that $P_{0}$ is on the line $P_{1}P_{2}$, and $P_{0}P_{3}{⊥P}_{1}P_{2}$, the following equation system can be established as
(59)$$\frac{x-x_{1}}{x_{2}-x_{1}}=\frac{y-y_{1}}{y_{2}-y_{1}}=\frac{z-z_{1}}{z_{2}-z_{1}}=m$$
(60)$$\left(x-x_{3}\right)\left(x_{2}-x_{1}\right)+\left(y-y_{3}\right)\left(y_{2}-y_{1}\right)+\left(z-z_{3}\right)\left(z_{2}-z_{1}\right)=0$$

Represent x, y and z in (59) by the formulas containing m, substitute them into (60), obtain m, and then we can obtain $\left(x,y,z\right)$. The above is the main idea of the source signal localization based on the perpendicular foot algorithm.

Overall, the similarities between the source signal localization algorithms based on the centroid and the perpendicular foot mentioned above lie in that: there are four base stations. First, the obstructed base stations are identified and their output data are removed. Then, the source signal is a near-field source for a certain base station. According to the algorithms above, a near-field source localization result can be obtained, which is a point $P_{3}$ in three-dimensional space. The source signal is a far-field source for the remaining other base stations. Find the common perpendicular lines of the two target observation lines starting from these base stations, and record the two perpendicular lines as $P_{1}$ and $P_{2}$, respectively. In this case, it is equivalent to obtaining two possible constraints for the position of the signal source, one near the point $P_{3}$, and the other on the common perpendicular line $P_{1}P_{2}$. So the final problem that needs to be solved is how to determine the final positioning result based on these two constraints. The difference between source signal localization algorithms based on the centroid algorithm and the perpendicular foot algorithm is that: The former takes the centroid of a triangle composed of three points $P_{1}$, $P_{2}$ and $P_{3}$ as the positioning result, which is equivalent to averaging the two constraint conditions; the latter takes the perpendicular foot as the positioning result, which is equivalent to selecting the position closest to the point $P_{3}$ while fully satisfying the second constraint condition. The performance of the above algorithms will be further verified through the following computer simulations.

## 4. Computer Simulations

A serial of simulations are carried out to verify the performance improvement of the proposed algorithms. In the simulations, we choose 3500 MHz as the signal frequency, which can be regarded as in the low frequency band of 6G. A UCA of 16-elements (M=16) with adjacent element spacing $d=\frac{\lambda }{3}\left(\lambda =0.0857\;m\right)$ is taken into consideration. The frequency band of 6G makes the size of such array not too large, so it is more feasible in practical application. What needs to be added is that the nonadjacent element spacing is greater than $\left(\lambda /2\right)$ in the preceding array.

Firstly, we consider about the experiment regarding the near-field source localization based on a single base station, which is used to evaluate the performance of the proposed methods based on short baselines and long baselines.

A near-field stationary signal is impinging on the array. The channel noise is supposed to be AWGN which is statistically independent from the source signal. The distance between the source and the origin of UCA is within (0.12 m, 0.50 m), which is in the Fresnel region. Combing with (32), the ambiguity number is $k_{0}=-2,-2,0,1,2$. $N_{s}=500$ Monte-Carlo runs are carried out to obtain the results. In the following, we first consider the condition for near-field source localization based on a single base station.

In the first experiment, the SNR and the snapshot number are set at 5 dB and 250. The azimuth angle is fixed at 60°. We evaluate the elevation estimation error when the elevation angle φ and the range r take some different values. Figure 5 shows the estimation error ∆φ maintains a small value which is less than 0.8°, regardless of the elevation angle and the range.

In the second experiment, we consider the situation where the azimuth angle and the elevation angle are fixed, then we checkout how the localization error changes as the range changes. The SNR and the snapshot number are set at 5 dB and 250. The signal is located at $\left(60{}^{\circ},40{}^{\circ},r\right)$, where r takes some values within the near-field region. Figure 6 shows that the proposed short-baseline method performs much better than the conventional 3-D MUSIC method within all Fresnel region. The proposed long-baseline method goes a step further in improving positioning accuracy compared with the short-baseline method, with an RMSE of localization less than 3 cm generally. In addition, the RMSE of localization does not change as the range changes, which coincides with the analysis result in (49).

In the third experiment, we examine the performance of the proposed algorithms and 3-D MUSIC algorithm versus the SNR which is varied from −10 dB to 15 dB. The snapshot number is set at 250. The signal is located at $\left(60{}^{\circ},40{}^{\circ},0.3\;m\right)$. Figure 7 shows the changing trend of positioning RMSE as the SNR changes. It is obvious in Figure 5 that the long-baseline method performs best while the traditional 3-D MUSIC method performs worst. And all the three methods can perform well in the high SNR region.

In the fourth experiment, we evaluate the range estimation performance of the above three methods versus the SNR which is varied from −10 dB to 15 dB. The snapshot number is set at 250. The signal is located at $\left(50{}^{\circ},50{}^{\circ},0.3\;m\right)$. Figure 8 shows the changing trend of the source distance estimation RMSE as the SNR changes. It can be found from Figure 8 that the long-baseline method performs best while the traditional 3-D MUSIC method performs worst. And when the SNR increases, the estimation error performances of all the methods can be improved evidently.

Secondly, we consider about the experiment regarding the near-field source localization based on multiple base stations, which is also used to evaluate the performance of the proposed methods based on perpendicular foot and centroid, respectively, and also compare with the single-base-station algorithm.

The following experiments are based on a 3500 MHz 6G low-frequency narrowband signal with a signal wavelength of 0.0857 m and additive white Gaussian noise. There are totally 4 base stations as shown in Figure 4. All base stations form a rectangular area of 2 m × 2 m, with coordinates of (0, 0, 0), (2, 0, 0), (2, 2, 0), and (0, 2, 0) in meters. All base station array structures are UCA with 16 antennas, with adjacent antenna spaced at one-third of the signal wavelength, which is 0.0287 m. Based on geometric relationships, the array radius can be calculated to be 0.0733 m. The signal in the 6G frequency band ensures that the array size is not too large, which is feasible in practical applications. The distance range of the near-field signal is calculated by the following equation as
(61)$$r\in \left[0.62{\left(\frac{D^{3}}{\lambda }\right)}^{\frac{1}{2}},\frac{{2D}^{2}}{\lambda }\right]$$

Therefore, the distance between the near-field source and the reference array element at the center of the array is $\lambda \in \left[0.12\;m,0.5\;m\right]$.

In the following experiments, parameter estimation and localization performance are evaluated by the RMSE of $N_{s}$ times Monte Carlo simulation experiments as
(62)$$RMSE=\sqrt{\frac{1}{N_{s}}\sum_{m=1}^{N_{s}}{\left({\hat{\beta }}_{m}-\beta \right)}^{2}}\;$$
(63)$$RMSE=\sqrt{\frac{1}{N_{s}}\sum_{m=1}^{N_{s}}\left[{\left({\hat{x}}_{m}-x\right)}^{2}+{\left({\hat{y}}_{m}-y\right)}^{2}+{\left({\hat{z}}_{m}-z\right)}^{2}\right]}$$
where β is the true value of the parameters to be estimated, ${\hat{\beta }}_{m}$ is the estimated value of the parameters in the *m*th Monte Carlo experiment. $\left({\hat{x}}_{m},{\hat{y}}_{m},{\hat{z}}_{m}\right)$ is the estimated source coordinates in the *m*th Monte Carlo experiment, and $\left(x,y,z\right)$ is the true value of the actual location coordinates of the source signal. In the following experiments, we set $N_{s}=100$.

*Simulation 1:* This experiment is used to evaluate the azimuth angle estimation performance of the proposed algorithm.

The position coordinates of the source signal is 0.2 m, 0.3 m, and 0.16 m, while its parameters related to base station A are: azimuth angle θ=56.31°, elevation angle φ=66.07°, distance r=0.3945 m<0.5 m, which can be regarded as a near-field source related to station A. The distances between the source signal and base stations B, C, and D are 1.8318 m, 2.4810 m, and 1.7192 m, respectively, so the source can be regarded as a far-field source related to these base stations. For the results using the single base station A, we use the near-field signal source estimation algorithm in Section 3.1. For the results using multiple base stations, the estimation algorithm proposed in Section 3.2.1 (centroid algorithm) is utilized. Figure 9 shows the RMSE of azimuth estimation for the two algorithms mentioned above when the SNR ranges from −10 dB to 15 dB.

From Figure 9, it can be seen that compared to the single-base-station algorithm, the multiple-base-stations estimation algorithm proposed in Section 3.2.1 (centroid algorithm) can effectively improve the estimation accuracy. The reason for this is: Firstly, it can identify and eliminate base station data with signal occlusion; and secondly it can try to utilize the estimation results of more base stations other than a single near-field base station. When the SNR is greater than 0 dB, the estimation error of azimuth tends to be stable, reaching a minimum of 0.02°. And for near-field sources, due to the short distance from the receiving array, the SNR is generally not too low, so the proposed algorithm often has high accuracy and excellent performance in near-field source scenarios.

*Simulation 2:* This experiment is used to evaluate the localization performances between the proposed algorithms based on a single base station and multiple stations.

The location of the source signal is 1.8 m, 1.9 m, and 0.35 m, while its parameters related to base station A are: azimuth angle θ=46.55°, elevation angle φ=82.38°. The distances from source signal to base stations A, B, C and D are 2.6173 m, 1.9423 m, 0.4153 m, and 1.8364 m, respectively, so it can be regarded as a near-field source related to base station C, and a far-field source related to other three base stations. We compare the localization performances of the following three methods: (1) a long-baseline phase interferometer with single-base-station localization; (2) a long-baseline phase interferometer with the perpendicular foot algorithm; (3) a long-baseline phase interferometer with the centroid algorithm. Figure 10 shows the positioning RMSE results (in meters) of the above three algorithms when the SNR ranges from −10 dB to 15 dB.

From Figure 10, it can be seen that compared to the single-base-station algorithm, the multiple-base-stations positioning algorithm proposed in this study can effectively improve the positioning accuracy because it effectively utilizes more measurement data and information from different base stations. In addition, when the same near-field source estimation algorithm is used for base station C, the error of the multiple-base-stations positioning algorithm based on the perpendicular foot method is slightly smaller than that of the multiple-base-stations positioning algorithm based on the centroid method, and the minimal error can even reach 0.002 m when SNR is 15 dB.

The following experiment is used to evaluate the localization performances comparison within the proposed algorithms with other state-of-the-art algorithms.

The location of the source signal is 0.2 m, 0.3 m, and 0.16 m, while its parameters related to base station A are: azimuth angle θ=56.31°, elevation angle φ=66.07°, distance r=0.3945 m. It can be regarded as a near-field source related to base station A, and a far-field source related to other three base stations. We compare the localization performances of the following five methods: (1) a long-baseline phase interferometer with the perpendicular foot algorithm; (2) a long-baseline phase interferometer with the centroid algorithm; (3) the conventional Root MUSIC algorithm considering the source signal as a far-field signal; (4) the path-following method [22] with the perpendicular foot algorithm; (5) the path-following method [22] with the centroid algorithm. And Figure 11 shows the positioning RMSE results (in meters) of the above five algorithms when the SNR ranges from −10 dB to 15 dB.

From Figure 11a, it can be observed first that the positioning accuracies of various algorithms are significantly improved as the SNR gradually increases. When all four base stations use the far-field Root MUSIC algorithm, the positioning error is significantly greater than the other four algorithms. The reason is that the signal models of the near-field signal source and the far-field signal source are different. When the far-field estimation method is still used for base station A, there will be a significant estimation error. Further observation of the positioning results of the other four algorithms in Figure 11b reveals that the RMSE of positioning errors can be ranked in the following descending order: the path-following method [22] with the centroid algorithm, the path-following method [22] with the perpendicular foot algorithm, a long-baseline phase interferometer with the centroid algorithm, a long-baseline phase interferometer with the perpendicular foot algorithm. That is to say, when base station A uses the long-baseline interferometer method proposed in Section 3 to estimate the near-field source position, regardless of whether the multiple-base-stations positioning uses the centroid algorithm or the perpendicular foot algorithm, the positioning error is smaller than the path-following method [22] used in base station A. So it reveals the advantages of the proposed algorithm based on a long-baseline interferometer again. In addition, when base station A uses the same near-field source signal estimation algorithm, the errors of the multiple-base-stations positioning algorithm based on the perpendicular foot algorithm and the multiple-base-stations positioning algorithm based on the centroid algorithm are relatively close, but the former one is slightly smaller than the latter one and has better performance, which may be a better choice for real applications.

## 5. Conclusions

Several 3-D near-field source localization methods are proposed in this study, which are based on the combination of a phase interferometer, the centroid algorithm and the perpendicular foot algorithm. The proposed approaches fit the applications under the single-base-station case and multiple-base-stations case, respectively. The corresponding computer simulations verify the effectiveness and the performance improvement compared with traditional methods.

## Figures and Tables

**Figure 1 sensors-24-06364-f001:**
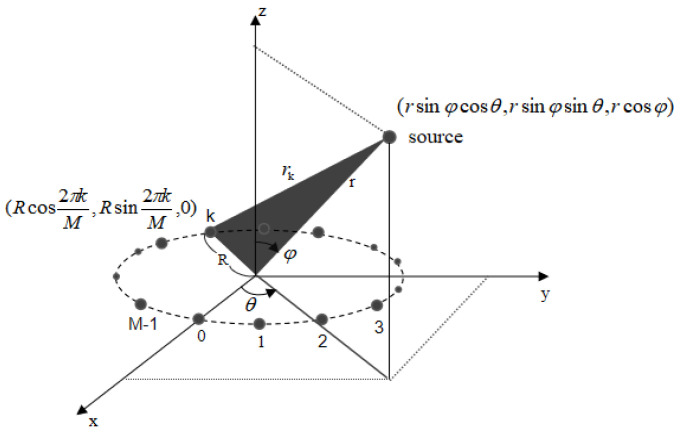
Geometry of the UCA.

**Figure 2 sensors-24-06364-f002:**
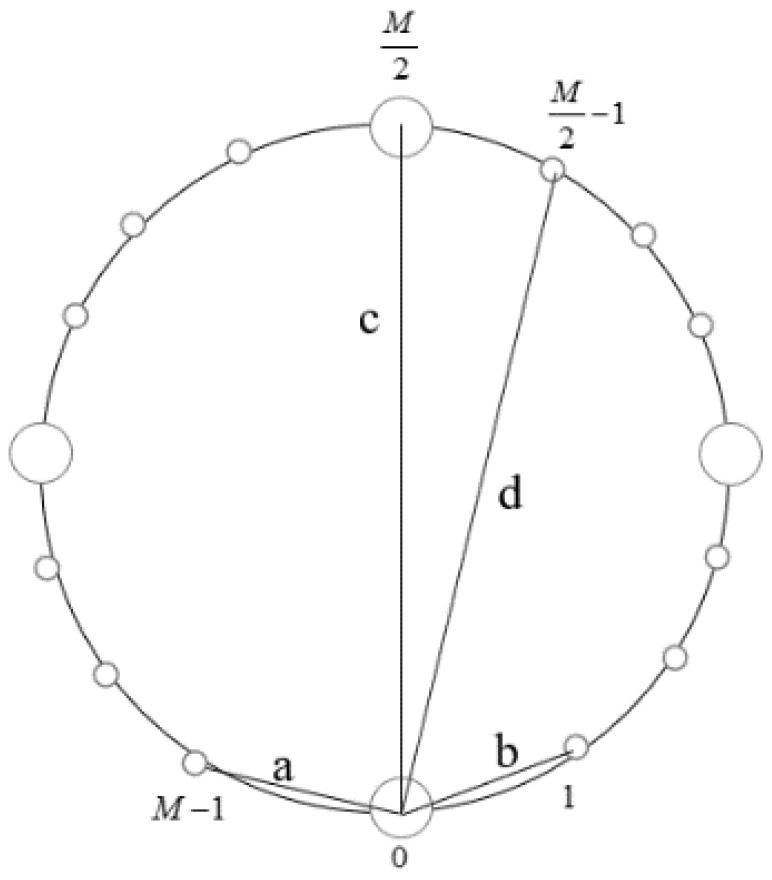
Geometry of baseline combinations. (a and b are short baselines, c and d are long baselines).

**Figure 3 sensors-24-06364-f003:**
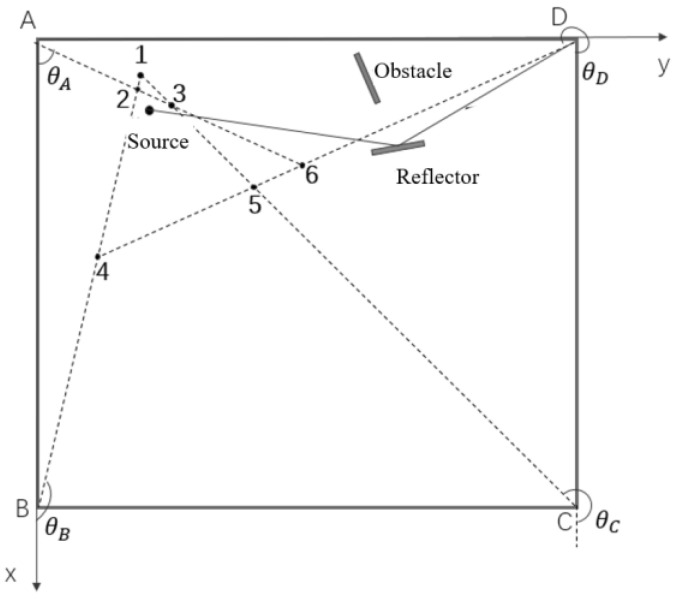
Example of multiple base stations and source signal. (A, B, C and D are base stations, and the numbers 1 to 6 are corresponding intersections of DOA estimations from each base station).

**Figure 4 sensors-24-06364-f004:**
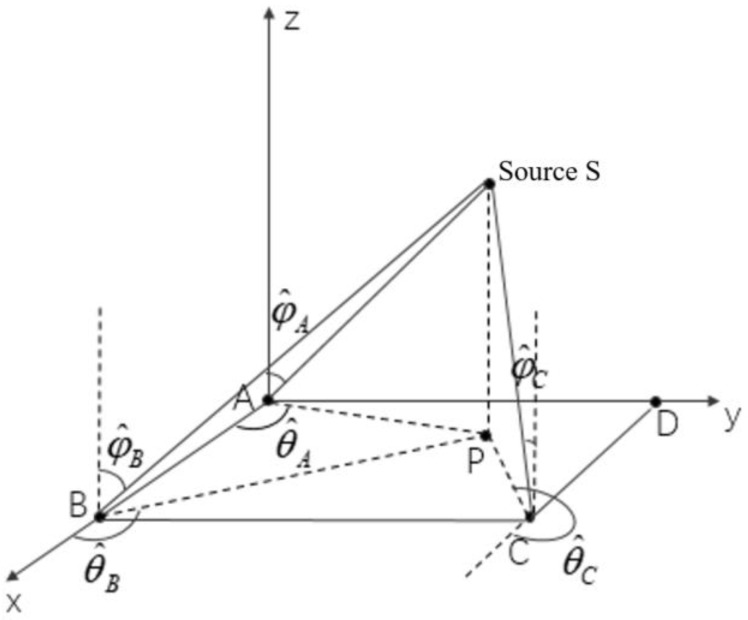
Location of the source signal in 3-D space. (A, B, C and D are base stations, P is the projection point of source S in the xoy plane, ${\hat{\theta }}_{i}$
$\left(i=A,B,C\right)$ are the estimated azimuth angles of three base stations A, B and C, and ${\hat{\varphi }}_{i}$ $\left(i=A,B,C\right)$ are the estimated elevation angles of three base stations A, B and C).

**Figure 5 sensors-24-06364-f005:**
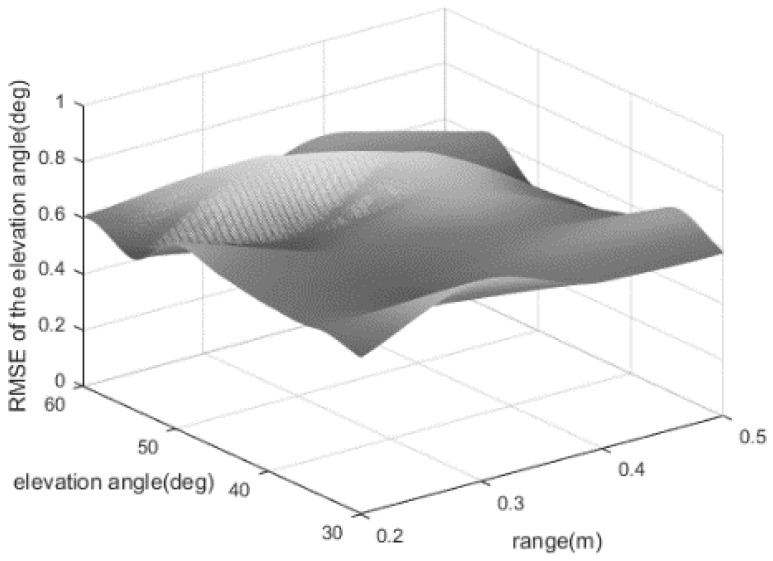
The RMSEs of the elevation angle for a near-field source.

**Figure 6 sensors-24-06364-f006:**
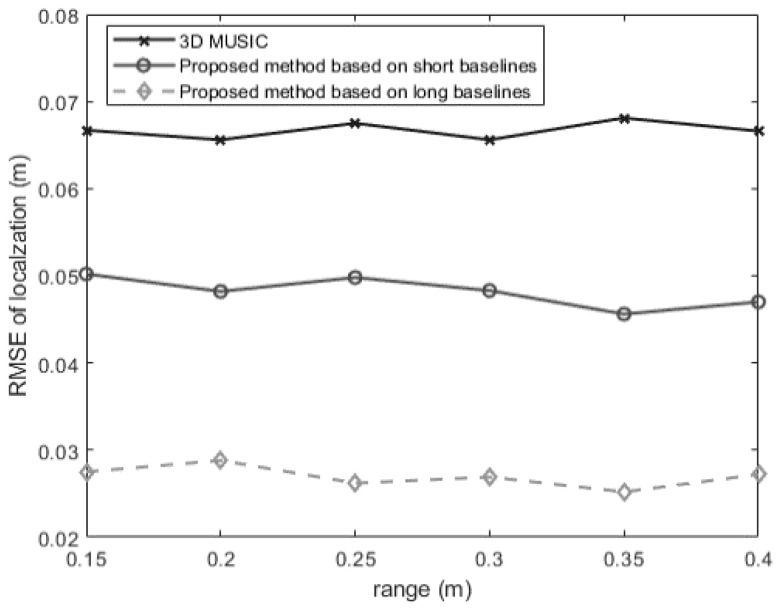
The RMSEs of localization for a near-field source versus the range (θ=60°, φ=40°).

**Figure 7 sensors-24-06364-f007:**
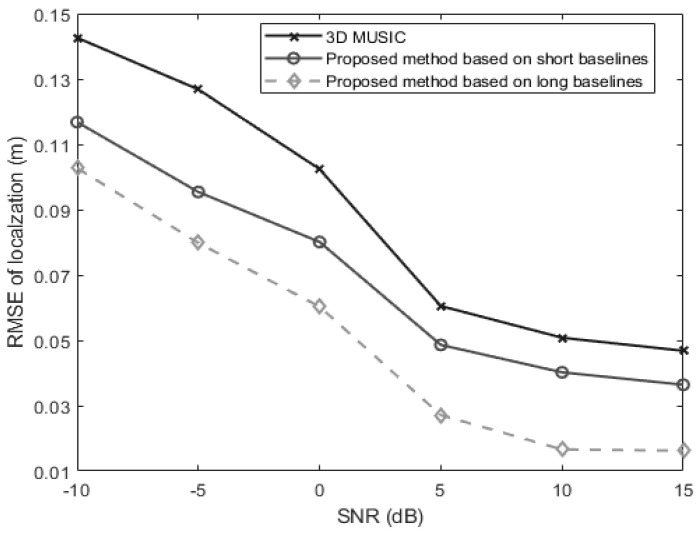
The RMSEs of localization for a near-field source versus the SNR (θ=60°, φ=40°, r=0.3 m).

**Figure 8 sensors-24-06364-f008:**
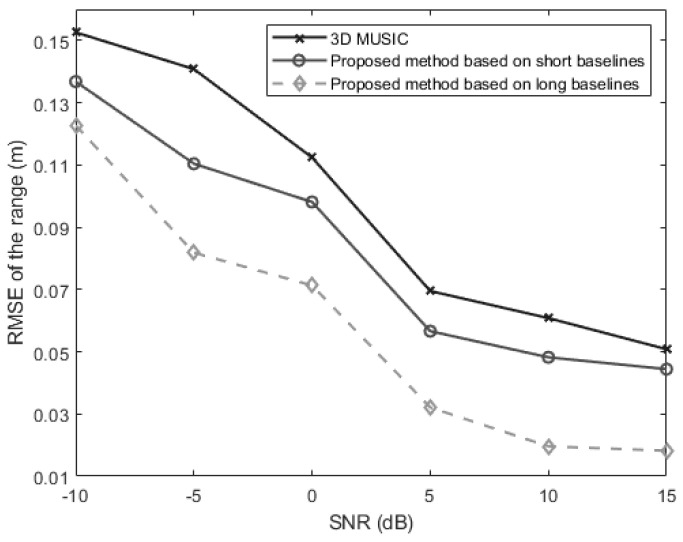
The RMSEs of the estimates of the range for a near-field source versus the SNR (θ=50°, φ=50°, r=0.3 m).

**Figure 9 sensors-24-06364-f009:**
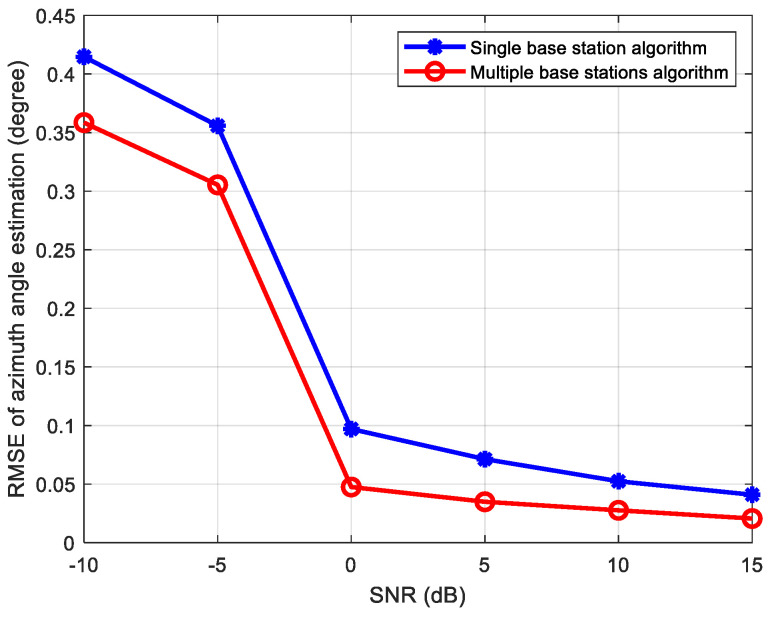
Comparison of azimuth estimation results.

**Figure 10 sensors-24-06364-f010:**
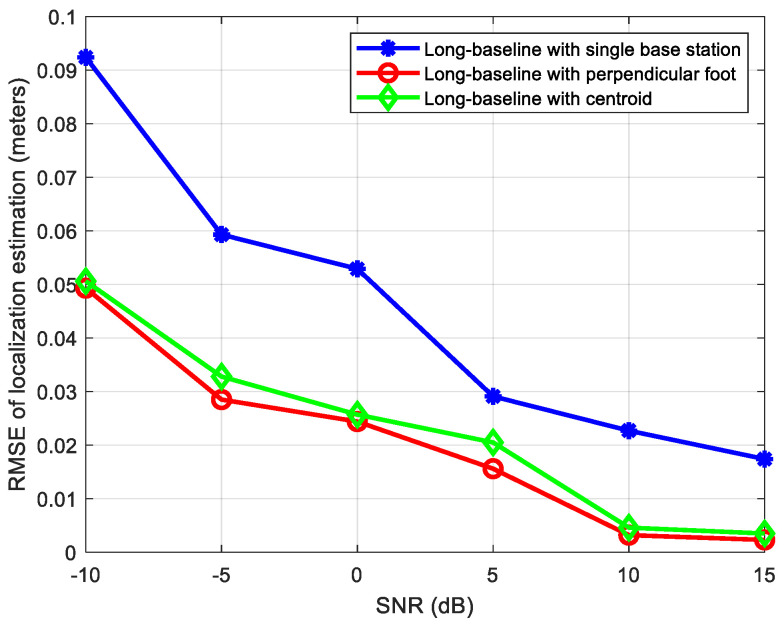
Positioning results comparison of multiple base stations and single base station.

**Figure 11 sensors-24-06364-f011:**
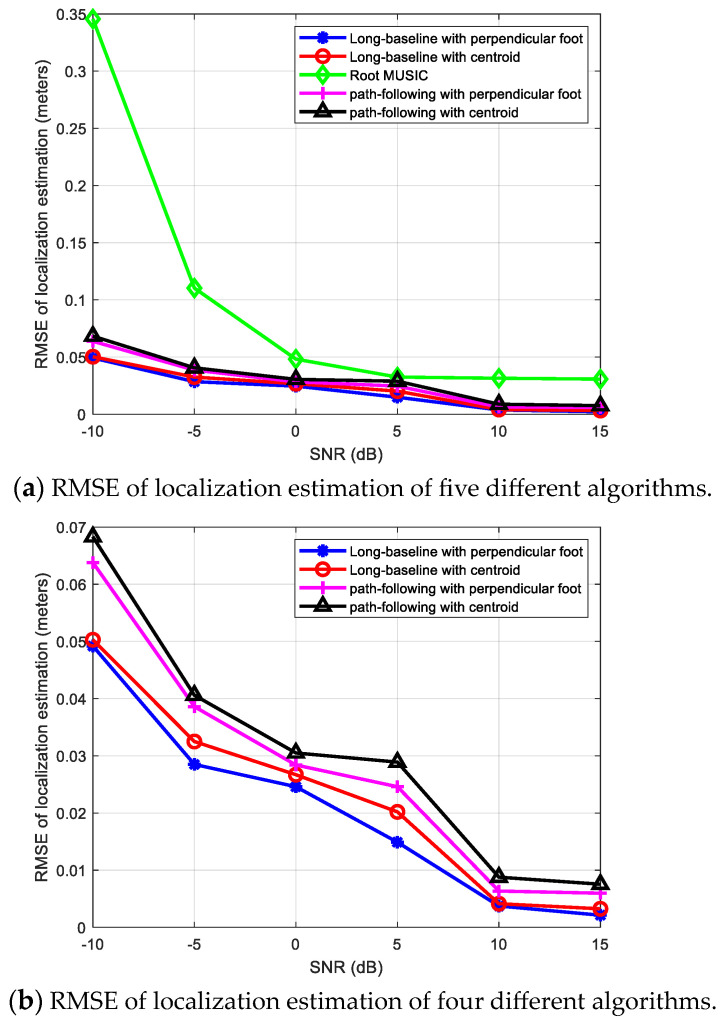
Positioning results comparison of five different algorithms.

## Data Availability

Data are contained within the article.

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
