# Peer review of "A 3-D Near-Field Source Localization Approach Based on the Combination of a Phase Interferometer, the Centroid Algorithm and the Perpendicular Foot Algorithm"

_sensors, 2024, doi:10.3390/s24196364_

Round 1

Reviewer 1 Report

Comments and Suggestions for Authors

This work proposes a 3D near-field source localization parameters estimation and localization algorithms for wireless near-field sources employing the uniform circular array structure. There are still some revisions and refinements to further improve readability and presentation.

1.     The paper has only a brief discussion on the relevant research background, there is a need to ensure that the proposed method is clearly innovative compared to the prior art. it will help the readers to judge the novelty of the work. The authors are suggested to improve the proposed work writeup in the introduction.

 2.    There are three experiments in this paper. It is suggested that the experimental description, experimental results and discussion analysis of the three experiments should be put together respectively to improve readability.

Reviewer 2 Report

Comments and Suggestions for Authors

This manuscript is devoted to development of several 3-dimensional parameters estimation and localization algorithms for wireless near-field  sources. Authors described the approaches used for determination of position of source of signal and corresponding algorithms developed by authors

The topic of manuscript is important for scientific and technical groups in areas of development of methods and approaches in wireless communication, radars etc.

There are some points to make the information more clear or to correct some details:

1.      It is necessary to explain the abbreviations at first mention. There are the abbreviation and without explanations (e.g., ESPRIT for possibly Estimation of Signal Parameters via Rotational Invariant Techniques (the first mention is in 64th line); EVD in 289th line, mMTC (the first mention is in 620th line); CSR (the first mention is in 621st line)).

2.      Authors wrote “A near-field  field source …”(92nd-93rd lines). It will be better to write the “field “one-time (e.g. “A near-field source” or simply  “A NF source…”.  Authors used the term of “source signal” (e.g. “whether the source signal is a near-field source or a far-field source” (310th line), position of the source signal” (406th line); 412th, 415th, 421st, 430th, 446th lines etc.). Here the authors are talking about the type of source of signal or position of source of signal but not signal. It must be corrected. The “source of signal” or “signal source” should be used.  

3.      It is not clear what is a in (13). Is it same A as in (4) (the letters must be same in this case) or other value (this fact should be described in this case)?

4.      It is not clear why the authors rewrite actually one equation (17) and (19) twice. These equations differ in notation (with  ‘ and without it). The same situation is with (18) and (20). If values with ‘ are measured values as  in 228th line, then  it should be corrected in 174th line (the values with ‘ should be there). But  in (17) and (18) the equations should be without ‘ and in (19) and (20) should be with ‘ (similarly to (35), (36)).

5.      The letters “y”[wai]  but not “g” (gamma) (as it should be )appear in (24) in two last terms (in 3rd power).

6.      Authors write the some notations of parts of text which are not used in this manuscript (“subsection III.A.3” (312th line),”section III” (608th line).It must be checked.

This manuscript is written sufficiently clear and describes with details the several 3-dimensional parameters estimation and localization algorithms for wireless near-field  sources developed by authors.

The manuscript can be published after minor revisions.

Comments on the Quality of English Language

The text should be checked. Authors used, e.g., the term of “source signal” (e.g. “whether the source signal is a near-field source or a far-field source” (310th line), position of the source signal” (406th line); 412th, 415th, 421st, 430th, 446th lines etc.). Here the authors are talking about the type of source of signal or position of source of signal but not signal. It must be corrected. The “source of signal” or “signal source” should be used. 

Round 2

Reviewer 1 Report

Comments and Suggestions for Authors

This work proposes a 3D near-field source localization parameters estimation and localization algorithms for wireless near-field sources employing the uniform circular array structure.The authors have been revised the introduction and some other issues in the paper according to the reviewer's suggestions.  I agree to accept this paper.